# The Impact of Duodenal Mucosal Vulnerability in the Development of Epigastric Pain Syndrome in Functional Dyspepsia

**DOI:** 10.3390/ijms232213947

**Published:** 2022-11-12

**Authors:** Tomoki Okata, Kiyotaka Asanuma, Kenichiro Nakagawa, Waku Hatta, Tomoyuki Koike, Akira Imatani, Atsushi Masamune

**Affiliations:** Division of Gastroenterology, Tohoku University Graduate School of Medicine, Sendai 980-8574, Miyagi, Japan

**Keywords:** acid, duodenal mucosa, epigastric pain syndrome, mucosal vulnerability, transepithelial electrical resistance

## Abstract

An unidentified cause of functional dyspepsia (FD) is closely associated with medication resistance. Acid suppression is a traditional and preferential method for the treatment of FD, but the efficacy of this treatment varies between epigastric pain syndrome (EPS) and postprandial syndrome (PDS): it is efficient in the former but not much in the latter. Transepithelial electrical resistance (TEER), a surrogate of mucosal barrier function, was measured under pH 3 and pH 5 acidic conditions using duodenal biopsy specimens obtained from the patients with EPS and PDS and asymptomatic healthy controls. The infiltration of inflammatory cells to the duodenal mucosa was accessed by immunohistochemical analysis. The duodenal mucosal TEER in EPS patients was decreased by exposure to the acidic solution compared to that of the controls and the PDS patients. The decrease in TEER of the EPS patients was observed even under pH 5 weak acidic condition and was correlated to degree of the epigastric pain. Moreover, the duodenal mucosa of EPS patients presented an increase in mast cells and plasma cells that expressed Ig-E. Duodenal mucosal vulnerability to acid is likely to develop EPS.

## 1. Introduction

Functional dyspepsia (FD) leads to persistent or recurrent bothersome symptoms that originate from the gastroduodenal region, despite the absence of readily identifiable organic disease [1]. FD occurs in up to 20% of the population, making it one of the most common gastrointestinal disorders, and impacts our social activity by impairing quality of life [2,3]. FD is thought to be caused by multiple mechanisms, including gastric sensorimotor dysfunction, alternation of gut–brain neural signaling, low-grade inflammation and impaired barrier function of the duodena mucosa, which could generate complexity of symptoms and resistance to treatments in the disease [4]. For effective strategies for patients’ management, the Rome criteria divides FD into two groups depending on primal symptoms: postprandial distress syndrome (PDS), which is mainly composed of meal-induced symptoms, such as early satiation or postprandial fulness, and epigastric pain syndrome (EPS), which is mainly composed of pain or burning in the epigastric region regardless of meals. However, the differences of pathologies of the two groups have yet to be fully elucidated [5].

Although the available treatment options are limited, gastric acid suppressive reagents have been administered preferentially for the treatment of FD [6]. Several studies have reported that both histamine-2 receptor antagonist (H2RA) and proton pump inhibitor (PPI) are effective in treating FD, despite not leading to complete remission [7,8]. In addition, acid suppressive treatment alleviates the symptom of EPS more effectively than the symptoms of PDS, indicating that functions of gastric acid in EPS are distinctly different from those in PDS with regard to the development of symptoms [9]. A potassium-competitive acid blocker (P-CAB), a new drug that has a pronounced function to reduce acid secretion compared to conventional acid suppressive reagents such as PPI and H2RA, is reported to be effective in FD [10]. Gastric acid is likely to be involved in the pathogenesis of FD, but the exact mechanism is still unknown.

Previous studies have reported that duodenal mucosal sensitivity to acid is related to the development of FD [11,12]. Acid injection into the duodenum in FD patients not only enhanced dyspeptic symptom in FD but also evoked epigastric pain [13,14]. Recently, FD has shown to be closely associated with low-grade inflammation of the duodenal mucosa, where infiltration of inflammatory cells, such as eosinophils and mast cells, were observed. The low-grade mucosal inflammation is likely to be correlated to impaired mucosal barrier function, which could be a key component in the pathogenesis of FD [15]. However, no study has elucidated whether duodenal mucosa exposure to acid is involved in an impairment of barrier function and mucosal inflammation and whether there are any differences between EPS and PDS patients. In the current study, we measured the transepithelial electrical resistance (TEER) in duodenal mucosa during acid exposure to clarify differences between EPS and PDS.

## 2. Results

### 2.1. Study Population

We enrolled 27 consecutive patients with FD who fulfilled the Rome IV criteria and 23 asymptomatic healthy controls (Table 1). The 27 FD patients consisted of 16 patients with EPS and 11 patients with PDS. The differences in gender, ages, body mass index (BMI) and smoking and drinking status did not reach significance between the patients and controls. In addition, eight of the FD patients and two of the controls had undergone HP eradication, and the difference in the degree of gastric atrophy did not reach statistical significance. All patients were administered acid suppressive reagents (rabeprazole or esomeprazole: *n* = 13, vonoprazan: *n* = 13, and famotidine: *n* = 1, *n*: patient’s number) with or without acotiamide (10 patients) or mosapride (2 patients), and no apparent difference in the contents of these medications were observed between the EPS and PDS subgroups. In the current study, the assessment of the FD symptoms using the gastrointestinal symptom rating scale (GSRS) [16] revealed abdominal pain-predominant symptoms in the patients with EPS and indigestion-predominant ones in those with PDS, while the total GSRS scores did not differ between the two groups (Table 2).

The gastric mucosal atrophy was evaluated by the classification of Kimura and Takemoto; none–mild (none, C-1 and C-2), moderate (C-3, O-1) or severe (O-2, O-3), BMI: body mass index, EPS: epigastric pain syndrome, PDS: postprandial distress syndrome, PPI: proton-pump inhibitor, P-cab: potassium-competitive acid blocker, SD: standard deviation, *n*: number, NS: not significant, NA: not available, a: Tukey–Kramer test among the 3 groups, b: Fisher’s exact test among the 3 groups, c: Fisher’s exact test for EPS vs. PDS.

### 2.2. Duodenal Mucosal Barrier Function against Acid Is Impaired in FD Patients with EPS

The basal TEER of duodenal biopsy samples were not different among the three groups in the current study (controls vs. EPS, controls vs. PDS, EPS vs. PDS: 2.8 ± 0.7 vs. 3.0 ± 1.0 Ωcm^2^; *p* = 0.31, 2.8 ± 0.7 vs. 2.8 ± 0.9 Ωcm^2^; *p* = 0.77, 3.0 ± 1.0 vs. 2.8 ± 0.9 Ωcm^2^; *p* = 0.38, Tukey–Kramer test, respectively). The exposure to pH 5.0 and pH 3.0 acidic solution decreased duodenal mucosal TEER in a time-dependent manner in all the groups (Figure 1). 

The exposure to a pH 3.0 solution decreased duodenal TEER compared to exposure to a pH 5.0 solution in only the patients with EPS, while the change of duodenal TEER in the exposure to a pH3.0 acidic solution was similar to that in the exposure to a pH 5.0 solution in the controls and the patients with PDS. Moreover, the treatments with acidic solution decreased the duodenal TEER in the patients with EPS compared to the controls, but there was no difference in the duodenal TEER between the patients with PDS and the controls (Figure 2). The difference between the average TEER of the patients with EPS and that of the controls reached a maximum at 90 min among other time points. Thus, we selected the TEER value of this time point for further assessment.

### 2.3. Symptoms in the Patients with EPS Are Related to Duodenal Mucosal Stimulation with A pH 5.0 Acidic Solution but Not a pH 3.0 Acidic Solution

The abdominal pain scores of the GSRS were moderately correlated with a reduction of duodenal TEER upon stimulation with a pH 5.0 acidic solution (correlation coefficient; r = 0.43). However, the stimulation with a pH 3.0 acidic solution did not cause any correlation between the abdominal pain scores and the change of the duodenal TEER (r = 0.005).

### 2.4. Duodenal Mucosa in Females Is Susceptible to Acid

Although clinical backgrounds such as gender, age, BMI and smoking and drinking status did not influence in duodenal TEER under the condition of stimulation with a pH 5.0 acidic solution, the female patients with EPS presented an apparent decrease in duodenal TEER compared to male patients under that with a pH 3.0 solution (*p* = 0.009, Student’s *t*-test). 

### 2.5. Development of EPS Is Related to Mucosal Infiltration of Mast Cells in the Duodenum

Patients with EPS presented increased mast cells in the duodenal mucosa among the three groups, while those with PDS presented a similar number of mast cells in the duodenal mucosa compared to the controls (Figure 3). Moreover, Ig-E co-stained CD138+ lymphocytes were also increased in the patients with EPS in the duodenal mucosa, while the number of CD138+ lymphocytes were not changed among the three groups (data not shown). In the current study, the increase in infiltration of eosinophils was not observed in the patients with FD compared to the controls. In addition, the number of mast cell in the duodenal mucosa was moderately correlated to the GSRS abdominal pain score in the patients with EPS (r = 0.51). In contrast, the number of Ig-E co-stained CD138+ lymphocytes did not present a correlation to the pain scores in the patients with EPS (data not shown).

## 3. Discussion

The current study is the first to demonstrate duodenal mucosal vulnerability to acid in the EPS patients even when taking acid-suppressive medication. Moreover, the duodenal mucosa in the EPS patients represented a distinctive pattern of inflammatory cells infiltrated into the mucosa. Acid-induced inflammation might be related to development of EPS in FD.

Acid-suppressive medications are administered preferentially to FD patients, but the treatment is often refractory in a clinical setting [5]. Although almost all the patients with FD recruited into the current study were administered either a PPI or P-CAB, the patients suffered from FD symptoms even when gastric acid secretion could be substantially reduced. The debate remains regarding the medications that can extensively suppress gastric acid. Thus, we conducted the current study without stopping taking the medication to explore drug-resistance factors in acid suppressive medication.

We found that impaired barrier function of the duodenal mucosa against acid was observed only in the patients with EPS, indicating that the development of EPS could be related to mucosal vulnerability to duodenal luminal acid, which could be associated with the difference between EPS and PDS regarding the mechanism of the disease development. This distinct mucosal reaction to acid in the duodenum could provide a pertinent reason why the efficacy of PPIs was evident in only EPS patients but not in PDS patients [8]. Moreover, we found that, in the patients with EPS, the duodenal mucosal TEER depended on the acidity the mucosa was exposed to and was decreased even in the pH5 weak acid stimulation, where the TEER was correlated to the GSRS abdominal pain score. Considering the daytime fluctuations of gastric pH during PPI and P-CAB administration [17], these findings suggest that exposing the duodenal mucosa to a weak acid could be important in the development of EPS. This finding could explain the reason why the use of a PPI failed to present better improvement of dyspeptic symptoms compared to H2RA [8]. The resistance to acid suppressive reagents in FD could have resulted from the duodenal mucosal susceptibility to weak acid in ESP patients and unresponsiveness to acid in PDS patients.

Low-grade inflammation in duodenal mucosal is characteristic in FD, which could play a pivotal role in the development of the disease [15]. The current study demonstrated the increased infiltration of mast cells in the duodenal mucosa of the EPS patients, which was accompanied by an increase in Ig-E-expressing plasma cells. Ig-E secreted from plasma cells can promote migration and activation of mast cells, leading to the release of chemical mediators that evoke inflammation [18]. These findings suggest that duodenal mucosal vulnerability to acid in patients with EPS could cause leaky epithelial barrier function, even to a weak acid, which facilitates the permeation of luminal causative antigens deeper into the mucosal layer, leading to the induction of a plasma cell-driven immune response and subsequent infiltration of mast cells, a putative key inflammatory cell in FD [19]. Contrary to the previous reports [15,20], eosinophils did not increase in the duodenal mucosa of the patients with FD compared to the controls. A recent study reported that PPIs inhibited the expression of eotaxin-3, an eosinophil chemotactic chemokine, in esophageal squamous cell lines, which could contribute to blocking of eosinophil recruitment into the mucosa [21]. Since P-CAB was reported to be effective in PPI-refractory eosinophilic esophagitis [22], P-CAB might inhibit eotaxin-3, which inhibited the increase in the duodenal mucosal eosinophils in the current study. This finding could raise another debate that duodenal eosinophils might not be responsible for the persistence of FD symptoms, at least with the administration of PPIs or P-CAB.

In the current study, a gender difference was found in duodenal mucosal barrier function regarding the pH 3 acidic solution. Considering that the patients in the current study were administered acid-suppressive reagents, female patients with EPS potentially had acid vulnerability in the duodenal mucosa, which is related to the female-predominant gender difference in FD [23,24]. Although a previous study reported that smoking and low BMI were related to the development of EPS [25], these clinical backgrounds did not present a correlation with the duodenal TEER under the condition of acid exposure. Moreover, patients with PDS experienced changes similar to the asymptomatic controls in TEER against acid exposure and no apparent difference in infiltration of inflammatory cells into duodenal mucosa compared to the controls, suggesting another mechanism might underlie in the pathophysiology of PDS.

In conclusion, different responses to acid in duodenal mucosa could underlie distinct pathologies between EPS and PDS, findings that might contribute to a strategy for adequate drug use suitable for the subgroups of FD.

## 4. Materials and Methods

### 4.1. Subject

Subjects were FD patients who met the Rome IV criteria and were diagnosed at Tohoku University Hospital from September 2017 to March 2020. As controls, nonsymptomatic healthy volunteers not taking any medication were enrolled. Written informed consent was obtained from all subjects prior to study entry. Information was collected from all subjects, including ages, gender, smoking and drinking status and body mass index (BMI), as well as a careful medical history, including *Helicobactor pylori* (HP) eradication therapy. Subjects with a smoking history were defined as those who had smoked any time prior to entry. Habitual drinkers were defined as those who drink alcohol 3 days or more per week. Any medications FD patients were currently being administered were not paused during examination in the current study. FD symptoms were scored using the GSRS. Subjects with peptic ulcer disease, a malignant disease, a history of previous esophagogastric surgery prior to the endoscopic examination or intake of nonsteroidal anti-inflammatory, corticosteroid, antiallergy or other immunosuppressive drugs were excluded from the current study. Patients with HP infection determined by the Urea breath test or serum HP antibody test were also excluded. The protocol was conducted according to the provisions of the Declaration of the Helsinki and approved by the Ethics Committee of Tohoku University Hospital (2018-2-178).

### 4.2. Endoscopic Examination

After regular endoscopic examination for systemic diseases causative of FD symptoms, the degree of gastric mucosal atrophy was evaluated using the classification of Kimura and Takemoto and was categorized into 3 groups [26]: none–mild (none, C-1 and C-2), moderate (C-3, O-1) or severe (O-2, O-3). Then, 3 duodenal mucosal biopsies 3–4 mm in size per patient were collected from the second part of the duodenum using a biopsy forceps, which had a 6.7 mm opening jaw size (No.10 Gaoke Third Road, Micro-Teck, Jiansu, China). For mini-Ussing chamber analysis, 2 biopsy specimens were immediately placed in ice-cold oxygenated modified Krebs buffer, and the remaining sample was fixed in formalin and embedded in paraffin for hematoxylin and eosin and immunohistochemical staining.

### 4.3. Electrophysiological Measurement of Transepithelial Electrical Resistance (TEER)

We used an electrophysiological measurement of TEER to evaluate duodenal mucosal barrier function using a mini-Ussing chamber system (EM-CYSY-2 Ussing Chamber Sytems, Physiologic Instruments, CA, USA) and a 2-channel voltage clamp with a preamplifier system (EVC4000-2, Physiologic Instruments), as previously reported [27]. In brief, duodenal biopsy samples were placed on a tissue mounting slide (Ussing Slider P2308, Physiologic Instruments) that has a 1 mm diameter circular aperture (area = 0.0079 cm^2^). The TEER was determined using Ohm’s law by passing a 10 μA current through the membrane and measuring the change in the transepithelial electrical potential difference. After acclimatization in pH 7.4 oxygenated Krebs buffer at 37 °C for 30 min, the TEERs were determined simultaneously at 2 biopsy samples with two different pH treatments every 15 min over a period of 120 min. A previous study demonstrated that the duodenal luminal acidity in FD patients was lower than that in healthy controls, reaching approximately pH 3 [11]. Moreover, the gastric pH fluctuates during the daytime and increases to approximately pH 5, even while taking P-CAB [17]. Therefore, pH 3 and pH 5 normal saline adjusted using hydrochloric were used as stimulations of the duodenal mucosa in the current study. The TEER value immediately prior to the acidic treatment was recorded as the basal value (time = 0). Values are expressed as percentages of change in resistance from basal value. For assessment of reproducibility, Ussing chamber results of 5 subjects from each group were studied. Every subject underwent the second TEER measurement in the same way as detailed above for the first experiment, and Cronbach’s α was calculated to assess the consistency of TEER between 2 measurements.

### 4.4. Immunohistochemistry

Paraffin-embedded tissue sections were cut into 4 μm slice and then deparaffinized and rehydrated with xylene and a graded alcohol series. The endogenous peroxidase activity of the samples was eliminated by immersion in 3% hydrogen peroxide in methanol for 10 min. The tissues were treated with a target retrieval solution (Dako, Glostrup, Denmark) in a pressure cooker at 125 °C for 5 min. Eosinophils, mast cells and plasma cells were incubated with mouse antieosinophilic major basic protein (MBP) (1:20; AbD Serotec, Kidlington, UK), rabbit antimast cell tryptase (1:200; Dako) and mouse antihuman CD138 (1:50; Dako) for plasma cells at 4 °C overnight incubation, respectively. For Ig-E staining, mouse anti-IgE antibody (1:100; Invitrogen) was used. Color development was performed using simple stain MAX-PO (Nichirei Bioscience) for MBP and tryptase and EnVision^TM^ FLEX (DAKO) for CD138 and IgE staining, followed by counterstaining with hematoxylin. The immune-positivity of Ig-E on plasma cells was determined by comparison to the serial section with CD138 staining. The immunostaining was quantified by counting on 3 randomly selected high-power fields for each sample (×400 magnification).

### 4.5. Statistics

All data are expressed as the mean ± standard deviation (SD). Student’s *t*-test was used for single comparisons, and the Tukey–Kramer test was used for multiple comparisons. Pearson’s correlation coefficient was applied to calculate correlations, and Fisher’s exact test was used to evaluate proportional differences. All analyses were conducted using R version 3.5.1 http://www.R-project.org/ (accessed on 10 November 2018). A *p* value < 0.05 was considered to indicate statistical significance.

## Figures and Tables

**Figure 1 ijms-23-13947-f001:**
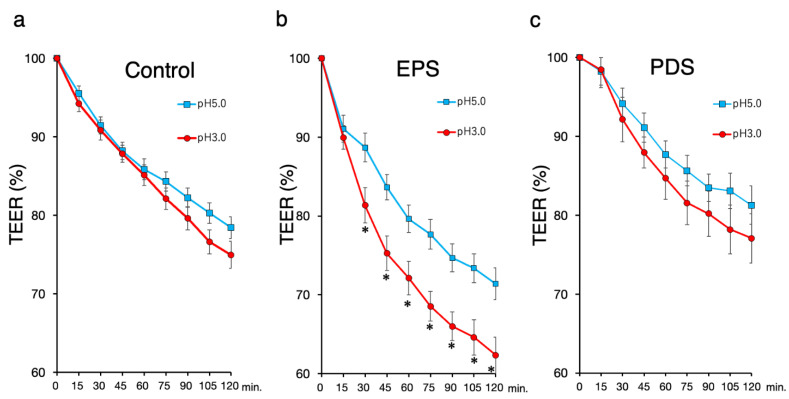
**Transepithelial electrical resistance (TEER) in the duodenal mucosa.** The exposure to a pH 3 or pH 5 acid solution was performed in each biopsy sample collected from the duodenal mucosa for 120 min during TEER measurements using a mini-Ussing chamber. Changes in the TEER are expressed as percentages relative to the initial values. Each plotted data point represents the mean ± standard deviation. (**a**) TEER in the controls (*n* = 23); (**b**) TEER in the EPS patients (*n* = 16); (**c**) TEER in the PDS patients (*n* = 11). Closed circle: the exposure to pH3 acidic solution, open square: the exposure to pH5 acidic solution. TEER: transepithelial electrical resistance, EPS: epigastric pain syndrome, PDS: postprandial distress syndrome. * *p* < 0.05, significant difference at each time point using Student’s *t*-test.

**Figure 2 ijms-23-13947-f002:**
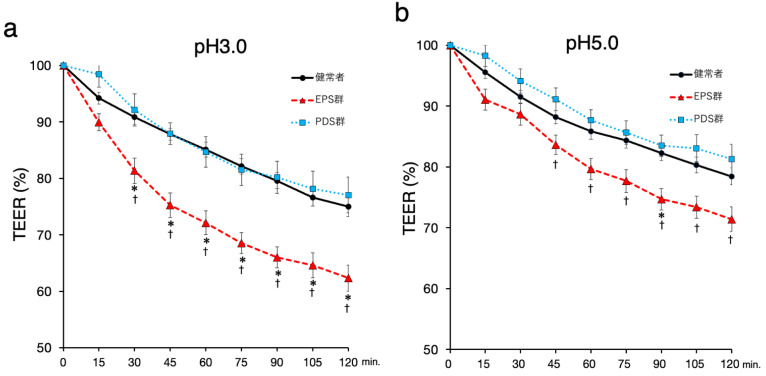
**The changes of duodenal TEER in the different acidity of the exposure.** Changes in the TEER are expressed as percentages relative to the initial values. Each plotted data point represents the mean ± standard deviation. (**a**) The duodenal TEER with the exposure to pH3 acid solution. (**b**) The duodenal TEER with the exposure to pH5 acid solution. Black circle and line: the healthy controls (*n* = 23), red triangle and dashed line: the EPS patients (*n* = 16), blue square and dotted line: the PDS patients (*n* = 11). TEER: transepithelial electrical resistance, EPS: epigastric pain syndrome, PDS: postprandial distress syndrome. *p* < 0.05, significant difference at each time point using Tukey–Kramer test; *, control vs. EPS; †, PDS vs. EPS.

**Figure 3 ijms-23-13947-f003:**
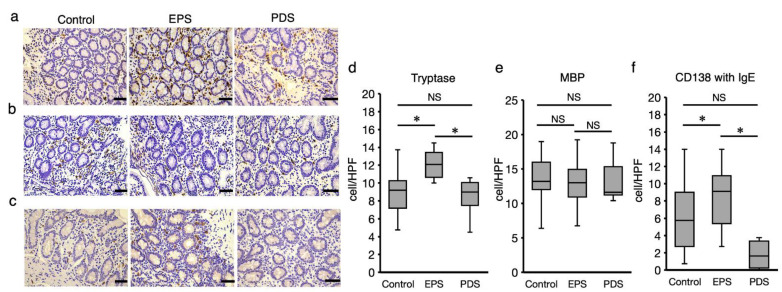
**Infiltration of immune cells in duodenal mucosa.** Duodenal biopsy specimens obtained from the EPS and PDS patients, and the healthy controls were stained immunohistochemically for mast cell using tryptase antibody (**a**), eosinophils using eosinophilic MBP antibody (**b**) and Ig-E in plasma cells using Ig-E (**c**). The immunostaining of each antibody was quantified by counting on 3 randomly selected high-power fields for each sample (×400 magnification) (**d**,**e**). The number of Ig-E-positive cells was counted within the cells that were stained with CD138, comparing with serial sections of each staining (**f**). The immune-positive cells were counted in 3 randomly selected high-power fields on each sample (controls: *n* = 23, EPS patients: *n* = 16, PDS patients: *n* = 11). EPS: epigastric pain syndrome, PDS: postprandial distress syndrome, MBP: major basic protein. * *p* < 0.05, Tukey–Kramer test, NS: not significant, Bar indicates 20 μm.

**Table 1 ijms-23-13947-t001:** Clinical background of the enrolled subjects.

	Control (*n* = 23)	EPS (*n* = 16)	PDS (*n* = 11)	*p*
Gender (male:female)	13:10	6:10	2:9	NS ^b^
Age (average ± SD)	51.7 ± 3.2	52.4 ± 3.8	54.3 ± 4.8	NS ^a^
BMI (average ± SD)	22.5 ± 0.6	22.0 ± 0.8	20.7 ± 0.7	NS ^a^
Smoke (no:yes)	20:3	13:3	9:2	NS ^b^
Drink (no:yes)	11:12	9:7	9:2	NS ^b^
Gastric atrophy (none~mild/moderate/severe)	21/2/0	11/3/2	8/2/1	NS ^b^
*H. pylori* (none:eradicated)	21:2	11:5	8:3	NS ^b^
Medication				
PPI or P-cab alone	NA	4	4	
PPI or P-cab + acotiamide	NA	9	5	NS ^c^
PPI or P-cab + others	NA	3	2	

**Table 2 ijms-23-13947-t002:** Gastrointestinal symptom rating score in the patients.

	EPS	PDS	*p*
abdominal pain	10.1 ± 1.0	5.7 ± 0.8	0.005
indigestion	9.3 ± 0.7	12.4 ± 1.7	0.08
reflux	5.4 ± 0.6	3.9 ± 0.5	0.09
constipation	6.4 ± 0.7	5.7 ± 0.9	0.57
diarrhea	4.9 ± 0.5	6.2 ± 0.8	0.12
total	36.2 ± 2.4	33.9 ± 2.9	0.55

Data are expressed as average ± standard deviation. EPS: epigastric pain syndrome, PDS: postprandial distress syndrome, Student’s *t*-test.

## Data Availability

The data presented in this study are available on request from the corresponding author. The data are not publicly available due to privacy.

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
