# Peer review of "The Impact of Duodenal Mucosal Vulnerability in the Development of Epigastric Pain Syndrome in Functional Dyspepsia"

_ijms, 2022, doi:10.3390/ijms232213947_

Round 1

Reviewer 1 Report

Dear Okata et al.,

Please find attached the draft manuscript with my comments. 

As general comments:

a. Quality of presentation of figures needs improvement.

b. Language of the writeup needs a review

c. Title needs rephrasing.

d. Very importantly the references needs a good review.

Regards

Reviewer 2 Report

transendothelial electrical resistance measures tight junction dynamics, and the study attempted to identify difference in TEER between EPS and PDS in FD. 

1. It seems that patients were allowed to continue any medications (PPI or P-CAB) they were taking during the time of biopsy. Wouldn't it be more accurate to discontinue medications prior to obtaining tissue samples in order to discern any alteration caused by medication? It is difficult to say that the results are solely determined by different pathophysiology of FD.

2. Just one minor comment. Method section is placed at the last part, which is probably the reason why "GSRS" appears for the first time in results section without explanation for what it stands for.

Round 2

Reviewer 2 Report

The authors revised the article according to comments made by the reviewers.